# Association between Disgust Sensitivity during Pregnancy and Endogenous Steroids: A Longitudinal Study

**DOI:** 10.3390/ijms25136857

**Published:** 2024-06-22

**Authors:** Šárka Kaňková, Daniela Dlouhá, Jana Ullmann, Marta Velíková, Josef Včelák, Martin Hill

**Affiliations:** 1Department of Philosophy and History of Science, Faculty of Science, Charles University, Viničná 7, 128 44 Prague, Czech Republic; sarka.kankova@natur.cuni.cz (Š.K.); dlouhada@natur.cuni.cz (D.D.); hlavajan@natur.cuni.cz (J.U.); 2Department of Steroids and Proteofactors, Institute of Endocrinology, Národní 8, 116 94 Prague, Czech Republic; mvelikova@endo.cz; 3Department of Molecular Endocrinology, Institute of Endocrinology, Národní 8, 116 94 Prague, Czech Republic; jvcelak@endo.cz

**Keywords:** steroids, disgust, pregnancy, behavioral immune system, testosterone, estrogens, androstenediol, DHEA, 7α/β-hydroxy-androgens, cortisol

## Abstract

The emotion of disgust protects individuals against pathogens, and it has been found to be elevated during pregnancy. Physiological mechanisms discussed in relation to these changes include immune markers and progesterone levels. This study aimed to assess the association between steroids and disgust sensitivity in pregnancy. Using a prospective longitudinal design, we analyzed blood serum steroid concentrations and measured disgust sensitivity via text-based questionnaires in a sample of 179 pregnant women during their first and third trimesters. We found positive correlations between disgust sensitivity and the levels of C19 steroids (including testosterone) and its precursors in the Δ^5^ pathway (androstenediol, DHEA, and their sulfates) and the Δ^4^ pathway (androstenedione). Additionally, positive correlations were observed with 5α/β-reduced C19 steroid metabolites in both trimesters. In the first trimester, disgust sensitivity was positively associated with 17-hydroxypregnanolone and with some estrogens. In the third trimester, positive associations were observed with cortisol and immunoprotective Δ^5^ C19 7α/β-hydroxy-steroids. Our findings show that disgust sensitivity is positively correlated with immunomodulatory steroids, and in the third trimester, with steroids which may be related to potential maternal-anxiety-related symptoms. This study highlights the complex relationship between hormonal changes and disgust sensitivity during pregnancy.

## 1. Introduction

In recent decades, there has been a growing interest in a concept at the intersection of psychology and immunology: the behavioral immune system. This system comprises a network of psychological mechanisms that serve as the first line of defense against potential pathogens from the environment. It refers to the series of adaptive behaviors that humans exhibit in response to various infectious threats. The affective part of the behavioral immune system consists of the emotion of disgust [1], which is linked to the avoidance of potentially harmful stimuli [2].

Regarding disgust sensitivity, one can observe both interindividual and intraindividual differences. It has been shown that disgust sensitivity changes during ontogenesis [3,4], and differences in disgust sensitivity have also been observed between men and women [2,5]. Moreover, disgust sensitivity can be influenced by numerous factors, including activity of the immune system [6,7], the presence of pathogens in the environment [8], reproductive status [9,10], and psychosocial influences [11,12]. Based on the premise of this variability in disgust sensitivity, the compensatory prophylaxis hypothesis has been proposed [13], assuming that the individual level of sensitivity to disgust is modulated depending on the current degree of immunosuppression.

Originally, the hypothesis was developed within the context of fluctuations in the levels of progesterone, a hormone believed to have immunosuppressive effects, and the consequent impact on immunosuppression during the menstrual cycle [14]. During the luteal phase, when progesterone levels peak, increased disgust sensitivity is believed to compensate for progesterone-associated immunosuppression. In subsequent years, numerous studies have tested the compensatory prophylaxis hypothesis in the context of the menstrual cycle and in relation to changes in the levels of progesterone (and other hormones). The reported findings did not, however, form a consistent pattern.

Some studies that investigated this hypothesis focused on changes in disgust sensitivity during the menstrual cycle, especially between the different phases of the cycle. The first cross-sectional study to investigate this hypothesis found no differences between the follicular and luteal phases of the cycle [13]. Similarly, subsequent cross-sectional studies found no support for the compensatory prophylaxis hypothesis. One of these studies used video stimuli to trigger disgust responses [15], while the other employed two different text-based questionnaires to measure disgust [16]. Additionally, two longitudinal studies, one of which again used video stimuli [17] and the other used a text-based questionnaire [18], also failed to find evidence supporting this hypothesis. The results of a cross-sectional study conducted by Rafiee et al. [19], which observed the relationship between estimated levels of progesterone and estradiol and pathogen-related disgust, also found no supporting evidence. On the other hand, the results of two different longitudinal studies do support the compensatory prophylaxis hypothesis: they found that women had higher disgust sensitivity during the luteal phase than during menstruation [20] or the follicular phase [21]. A cross-sectional study also found elevated disgust sensitivity during the luteal phase in a subsample of women who recently had an infection [22]. In a recent study, women in the luteal phase displayed a more negative attitude and higher sensitivity to disgust-related phrases compared to women tested during the follicular phase or menstruation [23].

Changes in disgust sensitivity during the menstrual cycle have also been observed in direct relation to hormone levels. The currently leading hypothesis claims that higher levels of progesterone are associated with higher disgust sensitivity. The first cross-sectional study found a positive association between the levels of salivary progesterone and disgust sensitivity to visual stimuli [24]. In a longitudinal study, progesterone levels in the serum positively correlated with disgust sensitivity but only during the mid-luteal phase [20]. While not directly measuring disgust sensitivity, two studies focused on the processing of disgusted facial expressions. Authors of the first study found a positive association between the levels of salivary progesterone and a higher tendency to perceive disgusted faces with averted gaze as more intense, which could signify an increased sensitivity to facial cues signaling a nearby presence of a pathogenic threat [25]. Another study found a negative association between salivary estradiol levels and the overall processing of disgusted faces with direct gaze, which is thought to express direct communication of disgust over violation of moral norms [26]. The authors suggest that progesterone and estradiol could modulate the perception of disgusted faces differently depending on the direction of gaze. There is some evidence for a link between progesterone and disgust sensitivity in animal models as well: a reappraisal of data from a recent study on mice by Kavaliers et al. [27] has shown that an injection of progesterone given to females increased disgust towards infected males (as measured via the frequency of females avoiding the odor of infected males) [28].

On the other hand, multiple studies did not find the expected association, including Timmers et al. [17], who found no association between changes in the levels of salivary progesterone and changes in self-reported disgust between the follicular and luteal phases of the cycle. A longitudinal study that measured salivary progesterone, estradiol, testosterone, and cortisol found no association between either of the hormones and disgust sensitivity [29]. Another recent longitudinal study likewise measured the levels of salivary progesterone, estradiol, testosterone, and cortisol and found no association with disgust sensitivity, neither in within-subject nor in between-subject analyses [18].

Simultaneously with the aforementioned research, Fessler et al. [9] extended the investigation of the compensatory prophylaxis hypothesis to pregnant women. They hypothesized that disgust sensitivity would be elevated during the first trimester of pregnancy; it would be consistent with older theories which assumed that women’s immune system is suppressed during this phase of pregnancy. While their assumptions were confirmed in both their cross-sectional study [9] and in another longitudinal study [30], it is essential to recognize how our understanding of immunosuppression in early pregnancy has evolved. Recent findings indicate that during early pregnancy, the maternal immune system undergoes complex immunomodulation, with some processes being suppressed and others (such as inflammatory processes) being elevated [31,32,33]. Despite that—and in accordance with the compensatory prophylaxis hypothesis—increased susceptibility to disgust was recently observed in early pregnancy in women with lower levels of certain cytokines, in women whose immune system is probably insufficiently activated [7], and in those who had lower maternal serum levels of free β-human chorionic gonadotropin (associated with pregnancy-induced immunotolerance) [34]. During the first trimester, a higher disgust sensitivity was also observed in women who reported recent health problems [10] or in association with increased concentrations of pathogens in the environment, such as during the COVID-19 pandemic [8]. Moreover, a recent study by Dlouhá et al. [16] showed a significantly higher disgust sensitivity in women during early pregnancy compared with non-pregnant childless controls, which indicates that higher disgust sensitivity in pregnancy may provide protection during a period that is sensitive to fetal neurodevelopmental disruptions.

Although two studies [9,30] found higher disgust sensitivity during the first trimester than in later pregnancy, Dlouhá et al. [10], in a recent longitudinal study, found increasing disgust sensitivity throughout pregnancy and even after birth. The authors of the study discussed a possible association with increasing progesterone during pregnancy [14], but progesterone levels decrease after childbirth, which is not consistent with the reported further increase in disgust sensitivity during the postpartum period. The increase might be attributed to a more intense need for protection against infections towards the end of pregnancy due to the approaching childbirth and subsequent care for the newborn. Another potential interpretation of these findings revolves around a positive correlation between disgust sensitivity and negative affectivity. Existing research indicates that disgust is linked to affective states such as phobias [35,36], depression [11], or anxiety [37] and that alterations in disgust levels coincide with changes in the symptoms associated with contamination-based obsessive-compulsive disorder (OCD) [12]. It is known that late pregnancy is associated with increased anxiety symptoms [38], and elevated fear of death has been observed in the third trimester [39] and during labor [40]. It has been shown that these factors, along with anxiety, are related to postpartum anxiety [41].

It is apparent from the above that the relation between hormone levels and changes in disgust sensitivity, especially in regularly cycling women, has been repeatedly discussed and that the results are inconsistent. On the other hand, in the context of pregnancy, although it is a period of significant hormonal and immunological changes, only one study to date has explored the influence of hormones on disgust sensitivity [34]. To better understand the association between hormonal changes and changes in disgust sensitivity and to shed more light on the proximal mechanisms of disgust regulation during pregnancy, the main aim of this study was to test a broad range of steroid hormones in the first trimester and then again in the third trimester of pregnancy and to analyze their relationships with simultaneously measured disgust sensitivity. We focused on associations between disgust sensitivity and the levels of hormones with immunomodulatory effects, such as progesterone, testosterone, cortisol, estradiol, or 7-oxygenated (7α-hydroxy, 7β-hydroxy, and 7-oxo) and 16α-hydroxy-derivatives of adrenal androgens dehydroepiandrosterone (DHEA) and 5-androstene-3β,17β-diol [42,43]. We have also carried out an explorative analysis regarding possible associations between disgust and the levels of other endogenous steroids. Previous research that investigated the circulating steroids accompanying various mental disturbances [43] suggests that the relationship between disgust levels, immunity, and psychological status may also be modulated by, for instance, estrogen levels, reduced sulfoconjugation of steroids, 7α-, 7β-, and 16α-hydroxy-metabolites of C19 Δ^5^ steroids, or 5α/β-reduced pregnane steroids.

## 2. Results

### 2.1. Descriptive Statistics

The final sample consisted of 179 women aged between 21 and 44 (Mean = 31.5, SD = 4.27), out of whom 109 (60.9%) were primiparous and 75 (41.9%) were pregnant with a male fetus. Most women had a university degree (78.8%) and were married (54.7%). A total of 8 (5%) women had hypertension, and 19 (11.9%) women had gestational diabetes mellitus. Out of the sample, 16 women (8.9%) reported that they smoked regularly before pregnancy and 18 women (10.1%) only occasionally. The levels of steroids quantified in the circulation of pregnant women in the first and third trimesters are reported in Table 1.

Disgust sensitivity was assessed via the Disgust Scale-Revised (DS-R) [44] and the Pathogen domain of the Three Domains of Disgust Scale (TDDS) [45] (for more details, see the Material and Methods section). The mean DS-R/TDDS scores and internal consistencies are shown in Table 2.

### 2.2. Association between Disgust Sensitivity and Steroid Levels in the First Trimester

We assessed the association between disgust sensitivity and steroid levels in the first trimester of pregnancy. In the orthogonal projections to latent structures (OPLS) model for the overall DS-R score, disgust was significantly predicted by a broad spectrum of steroids, such that higher levels of androstenediol, 17-hydroxypregnanolone, 5β-pregnane-3α,17α,20α-triol, 5α-androstane-3,17-dione, androsterone, androsterone sulfate, epiandrosterone, epiandrosterone sulfate, 5α-androstane-3α,17β-diol, 5α-androstane-3β,17β-diol, 5β-androstane-3α,17β-diol, and 3α,5β-tetrahydrocorticosterone and lower levels of estrone predicted higher overall DS-R score. This model explained 9.8% (6.8% after cross-validation) of the overall DS-R score variability (Table 3).

The model for the Core disgust subscale revealed that higher levels of androstenediol, testosterone, 5α-dihydrotestosterone, 5α-androstane-3,17-dione, androsterone, androsterone sulfate, epiandrosterone sulfate, 5α-androstane-3α,17β-diol, and conjugated 5α-androstane-3β,17β-diol and lower levels of conjugated pregnanolone predicted higher scores of Core disgust. This model explained 11.4% (9.7% after cross-validation) of the Core disgust score variability (Table 3, Figure 1).

The model for the Contamination disgust subscale revealed that higher subscale scores were predicted by higher levels of estrone, androsterone sulfate, epiandrosterone sulfate, etiocholanolone sulfate, epietiocholanolone sulfate, and conjugated 5α-androstane-3β,17β-diol. This model explained 8.2% (5.2% after cross-validation) of the Contamination disgust score variability (Table 3, Figure 1).

The model for the Animal reminder disgust subscale showed that the scores of this subscale were positively associated with the following steroids: testosterone, estrone, estradiol, 5β,20α-tetrahydroprogesterone, conjugated 5β-androstane-3α,17β-diol, and 5α-androstane-3,17-dione. This model explained 9.2% (7.4% after cross-validation) of the Animal reminder disgust score variability (Table 3, Figure 1).

In the OPLS model for the Pathogen disgust score of the TDDS, higher disgust was significantly predicted by higher levels of 5α-dihydrotestosterone, estradiol, 5β-pregnane-3α,17α,20α-triol, conjugated 5α-androstane-3β,17β-diol, and 5α-androstane-3,17-dione and lower levels of 5β,20α-tetrahydroprogesterone. This model explained 16.6% (13.5% after cross-validation) of the Pathogen disgust score variability (Table 4, Figure 1).

Regarding the covariates, no covariates (maternal age, maternal pre-pregnancy BMI, pregnancy length, maternal weight gain, parity, fetal sex, maternal diabetes and hypertension, maternal pre-pregnancy smoking) contributed to the explanation of the variability of the disgust scores measured by both the DS-R and the Pathogen domain of the TDDS.

### 2.3. Association between Disgust Sensitivity and Steroid Levels in the Third Trimester

We also assessed the association between disgust sensitivity and steroid levels in the third trimester of pregnancy. In the OPLS model for the overall DS-R score, disgust was significantly predicted by higher levels of 7α-hydroxy-DHEA, androstenediol, 5β-androstane-3α,17β-diol, and cortisol. This model explained 11.7% (8.8% after cross-validation) of the overall DS-R score variability (Table 5).

The model for the Core disgust subscale revealed that higher levels of pregnenolone, 17-hydroxypregnenolone, DHEA, 7α-hydroxy-DHEA, 7β-hydroxy-DHEA, androstenediol, 5α-androstane-3α,7α,17β-triol, 5α-dihydrotestosterone, epiandrosterone, and 5α-androstane-3β,17β-diol predicted higher scores of Core disgust. This model explained 11.7% (8.8% after cross-validation) of the Core disgust score variability (Table 5, Figure 2).

The model for the Contamination disgust subscale revealed that higher subscale scores were predicted by higher levels of pregnenolone, conjugated pregnanolone, conjugated epipregnanolone, conjugated 5α,20α-tetrahydroprogesterone, conjugated 5α-pregnane-3α,20α-diol, conjugated 5β,20α-tetrahydroprogesterone, conjugated 5β-pregnane-3β,20α-diol, and lower levels of testosterone and estradiol sulfate. This model explained 32.5% (23.1% after cross-validation) of the Contamination disgust score variability (Table 5, Figure 2).

The model for the Animal reminder disgust subscale showed that the scores of this subscale were positively associated with the following steroids: epiandrosterone, 5α-androstane-3β,17β-diol, 5β-androstane-3α,17β-diol, conjugated 5β-androstane-3α,17β-diol. This model explained 10% (7.8% after cross-validation) of the Animal reminder disgust score variability (Table 5, Figure 2).

In the OPLS model for the Pathogen disgust score of the TDDS, higher disgust was significantly predicted by higher levels of DHEA, DHEA sulfate, androstenediol, androsterone sulfate, epiandrosterone, epiandrosterone sulfate, and 5β-androstane-3α,17β-diol. This model explained 13% (11.1% after cross-validation) of the Pathogen disgust score variability (Table 6, Figure 2).

As for the covariates, only maternal age and parity contributed to the explanation of the variability of the Contamination disgust subscale score, such that older women, as well as multiparous women, reported higher levels of Contamination disgust.

### 2.4. The Effect of Changes in Steroid Levels during Pregnancy on Changes in Disgust Sensitivity

Finally, we assessed the association between the delta scores of disgust sensitivity (Δdisgust) and delta steroids (Δsteroids) calculated as the levels measured in the third trimester minus the levels measured in the first trimester. In the input models, the Δdisgust sensitivity was represented by vector Y. Matrix X was constituted by Δsteroids, steroid levels, and relevant disgust sensitivity measured in the first trimester of pregnancy (representing the baseline), as well as the related variables of maternal age, maternal BMI before pregnancy and ΔBMI, pregnancy length (both in the first trimester and Δ), maternal weight gain (both in the first trimester and Δ), parity, fetal sex, maternal diabetes and hypertension, and maternal pre-pregnancy smoking.

In the OPLS model for Δoverall DS-R score, higher Δdisgust was significantly predicted by higher values of ΔDHEA sulfate, Δandrostenediol, Δ5α-dihydrotestosterone, Δandrosterone, and Δ5α-androstane-3β,17β-diol, and also by lower levels of the overall DS-R score and by a broad spectrum of steroids, both measured in the first trimester (Table 7). As for the role of the covariates, older women and those who had a longer pregnancy at the time of the first trimester measurement were significantly positively associated with higher values of Δoverall DS-R score. This model explained 23.7% (19.3% after cross-validation) of Δoverall DS-R disgust score variability.

ΔCore disgust was significantly predicted by lower levels of overall DS-R score in the first trimester and by lower levels of steroids measured in the first trimester: 17-hydroxyprogesterone, 16α-hydroxyprogesterone, androstenedione, testosterone, 17-hydroxypregnanolone, 5α-androstane-3,17-dione, androsterone, androsterone sulfate, epiandrosterone sulfate, epietiocholanolone sulfate, 5α-androstane-3α,17β-diol, and 5α-androstane-3β,17β-diol. This model explained 12.4% (9.9% after cross-validation) of ΔCore disgust score variability (Table 7).

The model for ΔContamination disgust revealed that higher values of ΔContamination disgust were negatively associated with Δandrostenediol sulfate and also with many steroids measured in the first trimester (Table 7). Moreover, older women and those who had longer Δpregnancy length had higher ΔContamination disgust. The model for ΔContamination disgust explained 29.7% (24.9% after cross-validation) of ΔContamination disgust score variability.

In the model for ΔAnimal reminder disgust, higher Δdisgust was significantly predicted by higher values of Δ5α-dihydrotestosterone, Δ5α-androstane-3α,17β-diol, and Δpregnancy length, and again by lower levels of many steroids (e.g., testosterone, estradiol, androsterone) measured in the first trimester and also by lower levels of the Animal reminder disgust in the first trimester (Table 7). Moreover, women who had a longer pregnancy at the time of the first trimester measurement had significantly higher values of ΔAnimal reminder disgust. This model explained 19.2% (15.3% after cross-validation) of ΔAnimal reminder score variability.

In the OPLS model for ΔPathogen disgust score of the TDDS, higher Δdisgust was significantly predicted by higher values of pregnenolone (measured in the first trimester), Δestrone sulfate, Δestradiol sulfate, and Δestriol sulfate. Moreover, lower levels of Pathogen disgust measured in the first trimester significantly predicted higher ΔPathogen disgust. This model explained 14% (11.2% after cross-validation) of ΔPathogen disgust score variability (Table 8). No covariate was associated with ΔPathogen disgust.

## 3. Discussion

In order to better understand the physiological mechanisms involved in the changes in disgust sensitivity during pregnancy, we have investigated associations between a broad spectrum of steroids and disgust sensitivity in the first and third trimesters of pregnancy. In both trimesters, we found mainly positive correlations between disgust sensitivity and C19 steroids (androgens reflecting the activity of maternal adrenal *zona reticularis*), including the active androgens testosterone and 5α-dihydrotestosterone (both C19 Δ^4^ steroids). We have also observed associations between disgust sensitivity and androstenediol and DHEA (or their sulfates) as the precursors of active androgens in the Δ^5^ pathway, as well as androstenedione as a precursor in the Δ^4^ pathway. Although these positive associations were observed in both trimesters, they were significantly more pronounced in the first trimester for the C19 Δ^4^ steroids and in the third trimester for the C19 Δ^5^ steroids. In both trimesters, we have also observed positive correlations between disgust sensitivity and 5α/β-reduced metabolites of the C19 steroids.

In the first trimester, there was a positive association between disgust sensitivity, 17-hydroxypregnanolone, and estrogens such as estradiol and estrone (with the exception of the DS-R overall score, where we found a negative correlation with estrone). In addition to the substances mentioned above, we also found positive correlations between disgust sensitivity in the first trimester and the levels of certain C21 5α/β-reduced steroids, which are primarily derived from placental progesterone during pregnancy. In the third trimester, we found a positive association between disgust sensitivity and cortisol.

In line with our hypotheses, we therefore found that disgust sensitivity positively correlated with the levels of steroids that have immunomodulatory effects, such as testosterone, cortisol, estradiol, or 7α/β-hydroxy-, 7-oxo-derivatives of adrenal androgens, DHEA, and androstenediol (e.g., [42,43]). We have also confirmed the predicted association between disgust sensitivity and steroids such as estrogens, testosterone, cortisol, 7α-, 7β-, and 16α-hydroxy-metabolites of C19 Δ^5^ steroids, and 5α/β-reduced pregnane steroids, which were observed to be associated with mental wellbeing and certain mental disorders (e.g., [43,46]).

### 3.1. Immunomodulatory Steroids and Disgust Sensitivity in the First Trimester

Recent studies have reported that the first trimester is a time of significant immunomodulation [31,32]. Additionally, in line with the compensatory prophylaxis hypothesis [9,13], it has been demonstrated that disgust sensitivity negatively correlates with certain cytokines [7] and, similarly, that disgust sensitivity is elevated in the first trimester in women who were recently ill [10]. We therefore expected to observe significant relationships between disgust sensitivity and steroids with immunomodulatory function during this period of pregnancy as well. However, we did not find any significant correlations between disgust sensitivity and the most active immunomodulatory 7α/β-hydroxy-, 7-oxo-, and 16α-hydroxy-metabolites (e.g., 5-androstene-3β,16α, and17β-triol and its sulfate) of C19 Δ^5^ steroids in the first trimester. On the other hand, we observed a positive relationship between disgust sensitivity and the levels of androstenediol, a C19 Δ^5^ steroid, during this stage of pregnancy. Androstenediol, an adrenal androgen produced by the adrenal glands, serves as a precursor to key sex hormones such as testosterone and estrogen (both discussed separately below). Furthermore, androstenediol also plays a significant role in immunomodulatory processes [42,43].

We have also found positive correlations between disgust sensitivity in the first trimester and the levels of some C21 5α/β-reduced steroids, such as conjugated pregnanolone, 17-hydroxypregnanolone, 5α,20α-tetrahydroprogesterone, 5β,20α-tetrahydroprogesterone, 5β-pregnane-3α,17α,20α-triol, and 5α-androstane-3,17-dione. It is known that during pregnancy, the C21 steroids derive mainly from placental progesterone, which may be a product of LDL cholesterol penetrating from the maternal compartment into the placenta, but also from pregnenolone sulfate, which is formed in the fetal adrenal gland and then metabolized to progesterone in the placenta. These results thus suggest an important role of maternal and placental steroidogenesis in mother’s susceptibility to disgust.

In connection with the C21 5α/β-reduced steroids, 5α-androstane-3,17-dione deserves a separate mention. The levels of this steroid consistently positively correlated with disgust levels measured via the DS-R and TDDS questionnaires in the first trimester, as well as with disgust sensitivity measured via the DS-R in the third trimester of pregnancy. While not bioactive itself, 5α-androstane-3,17-dione is a precursor of 5α-androstanediols. Specific 5α-androstanols are known to act as positive GABAergic (neuroinhibitory) modulators, such as androsterone, etiocholanolone, and 5α-androstane-3α,17β-diol. Moreover, the biologically inactive 5α-androstane-3,17-dione is a direct 5α-reduced metabolite of the likewise biologically inactive androstenedione, which is a direct precursor of both the biologically active androgen testosterone and the biologically inactive estrone. Estrone, however, is further metabolized to create the active estrogen estradiol. In addition, the 5α-androstane-3,17-dione can be readily converted to the most active androgen 5α-dihydrotestosterone in a single metabolic step (via reduction of the oxo-group to 17beta-hydroxy-group), and this conversion works in both directions [47,48,49]. The presence of higher levels of 5α-androstane-3,17-dione can therefore indicate either a higher production of active sex steroids or, conversely, their higher catabolism. Considering that the results of our study also show a significant positive association between active sex steroids and disgust sensitivity, it would seem that in this case, the elevated levels of 5α-androstane-3,17-dione are more likely to be related to a higher production of these steroids and are thus in line with our prediction of the positive association between disgust sensitivity and sex steroids.

### 3.2. Immunomodulatory Steroids and Disgust Sensitivity in the Third Trimester

The associations between immunomodulatory steroids and disgust sensitivity were found not only in the first trimester but also in the third trimester. There we found positive correlations between cortisol, a steroid with well-known immunosuppressive effects [50], some steroids occurring in the metabolic pathway of cortisol synthesis, such as pregnenolone and 17-hydroxypregnenolone, and disgust sensitivity (measured via the DS-R).

Similarly, we also found positive correlations between Contamination disgust scores in the third trimester and multiple progesterone metabolites, such as conjugated pregnanolone, conjugated epipregnanolone, conjugated 5α,20α-tetrahydroprogesterone, conjugated 5α-pregnane-3α,20α-diol, conjugated 5β,20α-tetrahydroprogesterone, and conjugated 5β-pregnane-3β,20α-diol. These relationships may be related to the placental production of progesterone, although we observed no correlation with the hormone itself. Given that progesterone can be rapidly metabolized, these catabolites may be more stable markers of its presence, which could explain the absence of progesterone in these relationships. If this is indeed the case, it would be in line with the assumption of the compensatory prophylaxis hypothesis, according to which the immunosuppressive function of progesterone is compensated by elevated disgust sensitivity. This has been previously shown in two studies on non-pregnant, naturally cycling women [20,24] and in an animal model [28]. A positive association was also observed between progesterone and increased sensitivity to disgusted faces with averted gaze, which may signal a pathogen threat in the environment [25].

In addition to the progesterone metabolites and cortisol, our findings also showed a positive association between immunoreactive metabolites of C19 Δ^5^ androstanes and disgust sensitivity (measured via both the DS-R and TDDS). These associations were noted for DHEA, 7α-hydroxy-DHEA, 7β-hydroxy-DHEA, 5-Androstene-3β,7α,17β-triol, and again for androstenediol. These substances are known to not only stimulate the immune response but also to help suppress autoimmunity. While C19 Δ^5^ steroids and their metabolites can reduce the severity of autoimmune diseases [51,52,53,54,55,56], autoimmune diseases can, in turn, impair the production of adrenal C19 Δ^5^ steroids [51,57]. Additionally, some of these steroids may also counteract the suppression of the primary immune response by glucocorticoids [58]. It has also been reported that DHEA regulates the Th1/Th2 balance by either promoting the Th1 component or reducing the production of both components [55,59]. The C19 Δ^5^ steroids also suppress cell-mediated immunity and autoantibody formation [53,54,55,56,60], and they may induce restoration of the Th1-dominated cytokine profile.

The autoimmune response can also be triggered by estradiol through its interaction with estrogen receptors. This represents another mechanism of action for the C19 Δ^5^ steroids, which involves the catabolism of C19 estrogen precursors such as DHEA, androstenediol, and 5α-androstane-3β,17β-diol. These precursors, which are also estrogenic, are converted to their 7-oxygenated and 16α-hydroxylated catabolites. These metabolites cannot be further transformed into bioactive estrogens [61]. Interestingly, estradiol can stimulate catalytic steroid 7α-hydroxylase (CYP7B1) activity, mRNA, and human CYP7B1 reporter gene in human embryonic kidney cells HEK293. In turn, the stimulated catalytic CYP7B1 activity may control the DHEA and androstenediol levels in human tissues. These steroids serve as substrates for the synthesis of both, active androgens and estrogens [62]. The above mechanism could function as a negative feedback loop in the regulation of estrogen levels. Moreover, 5-androstene-3β,7β,17β-triol is immunoprotective despite its low concentrations and high clearance [63]. Synthetic anti-inflammatory derivatives of 5-androstene-3β,7β,17β-triol have been found to attenuate the production of inflammatory markers such as C-reactive protein, interleukin 17 (IL-17), TNFα, and interleukin 6 (IL-6), as well as to reduce the expression of mRNA for IL-6 and matrix metalloproteinase in inflamed tissues. Additionally, these steroids demonstrate suppressive effects on pro-inflammatory cytokines in the lungs and intensely stimulate splenic regulatory T-cells [64].

To summarize, the elevated levels of C19 Δ^5^ steroids and steroids such as cortisol, pregnenolone, or various progesterone metabolites during the third trimester of pregnancy could reflect an adaptive mechanism which would lead to increased protection against pathogens as childbirth approaches. After birth, the newborn is extremely vulnerable and since its own immune system is not yet developed, it relies primarily on maternal protection against infections.

### 3.3. Estrogens and Disgust Sensitivity

Focusing on results related to the first trimester of pregnancy, we have observed a positive correlation between disgust sensitivity (specifically the Contamination and Animal reminder subscales of the DS-R and the Pathogen domain of the TDDS) and estrogens (estrone and estradiol). The production of these estrogens during pregnancy relies on the production of C19 Δ^5^ steroid sulfates (DHEA sulfate, androstenediol sulfate) in the fetal zone of the adrenal gland [65]. This is also reflected in our results. In the first trimester, the levels of DHEA sulfate also positively correlated with the scores of the Contamination disgust subscale.

Similarly to the immunosuppressive effect of progesterone, which inspired the formulation of the compensatory prophylaxis hypothesis [13], estrogens, too, play a role in immunomodulation. They are known to shift the immune response towards Th2 dominance [66]. Moreover, some studies have observed higher levels of disgust sensitivity during the luteal phase of the menstrual cycle, when estrogen levels are elevated, compared to the menstrual phase [20,23], aligning with our findings. Conversely, our results are not in line with two studies that found no association between salivary estradiol levels and disgust sensitivity measured via TDDS [18,29]. However, it should be noted that both of the aforementioned studies were conducted on a population of non-pregnant women, and they focused on the relationship between disgust and estradiol levels during the menstrual cycle. Moreover, the hormone levels were measured in saliva samples in both studies, as opposed to blood serum, which was used in our study.

Nevertheless, our results regarding estrone levels are also not entirely unambiguous. While higher estrone levels were associated with higher scores of the Contamination and Animal reminder disgust subscales and the Pathogen disgust domain, the overall DS-R score correlated negatively with estrone levels. It must be taken into consideration that aside from the Contamination and Animal reminder disgust subscales, the DS-R also contains a Core disgust subscale, focused on food and animal or bodily products, which also contributes to the overall score. This subscale may have been the cause of the observed reverse direction of correlation. A better understanding of the observed effects would, however, require further research.

### 3.4. Testosterone and Disgust Sensitivity

A significant positive correlation with testosterone levels was observed only in the first trimester, with the sole exception of the Contamination disgust subscale score, which increased in association with decreasing testosterone levels (this relationship is discussed below in connection with specific anxiety-related outcomes). In both trimesters, we have also found a significant positive correlation between disgust sensitivity and 5α-dihydrotestosterone, which is directly produced from testosterone via the enzyme 5α-reductase in peripheral tissue.

Since higher testosterone levels are associated with increased immunosuppression [67], our main findings are in line with the compensatory prophylaxis hypothesis [13], suggesting that disgust sensitivity should be elevated during immunosuppression. Moreover, previous studies have observed higher disgust sensitivity in women pregnant with male compared to female fetus [10,30], possibly reflecting mechanisms that lead to elevated testosterone levels in women pregnant with a male fetus [68]. However, studies examining the relationships between salivary testosterone levels and disgust sensitivity in non-pregnant female population samples found no significant associations [18,29]. Additionally, our study found no evidence that the sex of the fetus influences disgust sensitivity during pregnancy, as fetus sex did not statistically significantly contribute to any of the analyses.

### 3.5. Steroids, Anxiety-Related Disorders, and Disgust Sensitivity in the Third Trimester

In the third trimester, we found a positive association between disgust sensitivity and cortisol and DHEA. This could be related to anxiety-related disorders, including OCD, as higher levels of cortisol and DHEA have been observed in female patients with OCD compared to a control group of women [46]. Moreover, this observation aligns with the positive association between disgust and anxiety-related disorders, including OCD [12,69,70], and could also help explain the specific results regarding the relationship between testosterone and the Contamination disgust subscale of the DS-R questionnaire.

In contrast to all other findings regarding disgust sensitivity and steroids, a lower level of testosterone in the third trimester predicted higher disgust scores in the Contamination subscale. It is also important to note that the testosterone levels had the strongest effect in this particular model: they explained 32.5% of variability. Similar to cortisol and DHEA, which were previously associated with anxiety disorders, lower salivary testosterone levels were measured in women with current depressive disorder, generalized anxiety disorder, social phobia, and agoraphobia without panic disorder [71].

That indicates that the increased disgust sensitivity during pregnancy might not only be an adaptive mechanism aimed at protecting the organism from pathogens. In the third trimester, it may also reflect the higher anxiety observed during this period [38,72], which may be associated with the approaching childbirth. The Contamination disgust subscale of the DS-R is centered around worries about the interpersonal transmission of pathogens and subsequent aversion. Contamination disgust is closely related to some types of OCD, whose symptoms include compulsive cleaning and handwashing. Our findings are also consistent with the study by Dlouhá et al. [10], where the authors observed increasing levels of disgust sensitivity during pregnancy that extended into the postpartum period.

Our findings underscore the importance of monitoring hormonal and psychological health in women in the third trimester of pregnancy to mitigate potential anxiety and OCD symptoms. Given that increased disgust sensitivity during pregnancy persists into the postpartum period, the observed associations may have long-term consequences for women’s mental health after childbirth.

### 3.6. The Effect of Changes in Steroid Levels during Pregnancy on Changes in Disgust Sensitivity

Regarding the correlations between changes in disgust sensitivity measured via DS-R and increasing pregnancy length, there was a general trend towards negative correlations between increasing disgust sensitivity and the overall activity of steroidogenesis during the first trimester of pregnancy. We have also observed positive correlations between disgust sensitivity and the increase in the C19 steroid levels between the first and third trimesters. This shows that on the one hand, lower steroidogenic activity in the first trimester is associated with a more significant increase in disgust sensitivity with increasing pregnancy length due to increased steroid production; on the other hand, a positive association between *zona reticularis* activity in the maternal adrenal gland and disgust sensitivity was also found. These data indicate that the role of the fetus is not decisive here because although the fetal zone of the fetal adrenal gland produces even more sulfated Δ^5^ androstanes than the *zona reticularis* in the maternal adrenal gland, these substances are rapidly metabolized to estrogens in the placenta, so while maternal blood estrogen levels increase exponentially with advancing gestational age, maternal adrenal androgen levels do not [65]. However, positive correlations between the increase in the scores of the Pathogen disgust domain of TDDS and the increase in estrogen sulfates also suggest an association with the activity of the fetal zone of the fetal adrenal gland. Interestingly, this zone is a fetal counterpart of maternal *zona reticularis*.

### 3.7. A Comparison of Questionnaires Measuring Disgust

The associations between specific steroids and the two different questionnaires, the DS-R and the Pathogen domain of TDDS, allow us to make some observations about the questionnaires themselves. The Pathogen domain of TDDS is often associated with similar steroids as the Core and Animal reminder subscales of the DS-R. During the first trimester, only a minimum of steroids is specifically associated only with the Pathogen domain, which suggests that the Pathogen domain reflects some part of the disgust sensitivity measured via the DS-R. The DS-R has been previously criticized for not effectively reflecting the adaptive function of disgust. Based on this critique, the TDDS questionnaire was developed to focus specifically on the adaptiveness of disgust sensitivity [45]. The results obtained from the two different questionnaires would suggest that the DS-R actually reflects both the adaptive function of disgust and the maladaptive form of disgust associated with, for instance, anxiety disorders.

### 3.8. Strengths and Limitations

Our study has several strengths. Due to its longitudinal design, we were able to track the development of disgust levels in response to changes in steroid levels during pregnancy. Moreover, we had a relatively large sample of women, which is rather uncommon for this type of study. Another strength of the study is the simultaneous use of two of the most commonly used textual questionnaires (DS-R and TDDS) to measure disgust sensitivity. The inclusion of both questionnaires enabled us to better understand the various aspects of disgust and to detect overlaps and differences between the questionnaires.

The main limitation of this study is that disgust sensitivity was based on self-report textual questionnaires. Aside from textual questionnaires, there are other methods of measuring disgust sensitivity, such as those based on visual stimuli. Such methods can also measure the experienced emotion based on various physiological parameters that directly reflect the subject’s state. In the case of pregnant women, however, we could not use such methods for ethical reasons. Nevertheless, some studies used the textual and visual methods in parallel and found no differences in the results acquired by the two approaches [16].

## 4. Materials and Methods

In a prospective longitudinal study running between June 2019 and November 2022, we collected data from pregnant women in the first and third trimesters of pregnancy. This study was part of a larger project to explore longitudinal changes in pregnancy and their correlations with biological and psychological factors.

### 4.1. Procedure—Data Collection

In total, 228 adult women who conceived naturally and reported no severe chronic diseases or autoimmune disorders were recruited for the study in collaboration with three gynecological clinics in Prague, Czech Republic. Most women were from the ProfiGyn clinic (*n* = 133), 37 women from the GynFleur clinic, and only nine women were recruited in the Levret clinic. From this sample, 49 women were excluded from the study: 15 women miscarried, 19 women either did not provide blood samples in both trimesters or there was an insufficient amount of blood serum for steroid hormone testing, and 15 women left the study at their request. The final sample thus consisted of 179 women.

Participants were recruited for the study during their first antenatal medical checkup during which their pregnancy was confirmed by their gynecologist. At this time, between weeks 5 and 14 of pregnancy (mean ± SD = 7.7 ± 1.23), they completed a background questionnaire that included questions about age, physical parameters, parity, the method of conception, health status, and several demographic questions. During medical checkups in the first trimester, between weeks 9 and 14 of pregnancy (mean ± SD = 10.3 ± 0.91), and again during the third trimester, between weeks 30 and 38 pregnancy (mean ± SD = 33.0 ± 1.61), they completed questionnaires that measured disgust sensitivity and provided blood samples for determining the levels of steroid hormones. Information about the sex of the baby was obtained from both medical records and questionnaires.

All women participating in this study were part of a larger study focused on prenatal factors that affect the mother and child’s wellbeing and health. Participants answered all the disgust questions along with pregnancy nausea questions and were not explicitly informed that the study was about disgust sensitivity because that could influence their responses. All women signed an informed consent form and participated in all parts of the study under a pseudo-anonymous code. The project was approved by the Institutional Review Board of the Faculty of Science, Charles University (Approval No. 2018/6 and 2019/10). All methods were performed in accordance with the relevant guidelines and regulations.

### 4.2. Questionnaires

All women completed a background questionnaire, including information about age, history of previous pregnancies, education level, size of residence area, and health data. Disgust sensitivity was assessed via the Disgust Scale-Revised (DS-R) [44] and via the Pathogen domain of the Three Domains of Disgust Scale (TDDS) [45].

The DS-R [44] is a self-report questionnaire containing 25 items, which are divided into three subscales: 12 items in the Core disgust subscale (disgust evoked by food and animal or bodily products); 8 items in the Animal reminder disgust subscale (disgust related to injuries, mortality, or violations of the body envelope); and 5 items in the Contamination disgust subscale (disgust associated with transmission of pathogens between people). The questionnaire has two parts: in the first part, participants rate to what extent they agree with presented statements on a scale from 0 = “Strongly disagree (very untrue about me)” to 4 = “Strongly agree (very true about me)”; in the second part, the participants rate how disgusting they would find situations described in the presented statements, again on a scale from 0 to 48 in the Core subscale, from 0 to 32 in the Animal reminder subscale, and from 0 to 20 in the Contamination subscale. The overall score can range from 0 to 100, with a higher score indicating higher disgust sensitivity. In cases where one-fifth or fewer questions were left unanswered within a subscale, we used the average score of that subscale to substitute the missing values (we substituted seven responses in the first trimester and five responses in the third trimester). In cases where more than one-fifth of answers were missing, the data from that participant were excluded from further analyses (ten in the first trimester and three in the third trimester).

The TDDS [45] is a 21-item self-report questionnaire containing 21 items, divided into three domains: Pathogen, Moral, and Sexual (each containing 7 items). For this study, only the Pathogen domain was used, as it was the most relevant to the aims of the study. In this questionnaire, the participants rate possibly disgusting statements on a scale from 0 (not disgusting at all) to 6 (extremely disgusting). The final score for this domain can range from 0 to 42. Once again, if less than one-fifth of the questions were left unanswered, the average score of the domain was used to substitute the missing values (one response in the first trimester and two responses in the third trimester). If more than one-fifth of the questions were missing, the data of that participant were excluded from further analyses (five in the first trimester and three in the third trimester).

### 4.3. Laboratory Measurement of Steroid Hormones

Blood samples of 179 women (the final sample) in their first and third trimester of pregnancy were analyzed for concentrations of steroid hormones in blood serum. Serum from blood was obtained after centrifugation (2 min at 3000× *g* at 21 °C) and stored at −20 °C until analyzed. We used the advanced gas chromatography trandem mass spectrometry (GC-MS/MS) platform for the multicomponent quantification of endogenous steroids: 56 unconjugated steroids and 35 polar conjugates of steroids (after hydrolysis). Unlike the current methods used for the quantification of circulating steroids on the GC-MS/MS platform, the present one was validated not only for the blood of men and non-pregnant women but also for the blood of pregnant women and for mixed umbilical cord blood [73]. The spectrum of analytes includes common hormones operating via nuclear receptors as well as other bioactive substances like immunomodulatory and neuroactive steroids. The present method was extended for corticoids and 17-hydroxylated 5α/β-reduced pregnanes, which are useful for the investigation of an alternative “backdoor” pathway. The testing was carried out in the Institute of Endocrinology under the direction of Martin Hill, PhD, DSc.

### 4.4. Statistics

At first, the data were both manually and automatically controlled. Using STATGRAPHICS Centurion 18 software (The Plains, VA, USA), all variables, with the exception for binary variables, were transformed towards symmetric distribution and constant variance before the final statistical analyses. Power transformations were used to transform the values of all analyzed variables, including steroids, to reach symmetric data distribution and homoscedasticity. Using the SIMCA software (v. 12.1.1.1), these transformed data were then automatically converted to z-scores for further analyses.

Multivariate regression with a reduction of dimensionality, known as orthogonal projections to latent structures (OPLS) [74], was conducted to assess the associations between hormonal changes and changes in disgust sensitivity during pregnancy. The OPLS model searches for the best linear combination of predictors for an optimum estimate of the dependent variable. Statistical software SIMCA v. 12.1.1.1. (Umetrics, Umea, Sweden) was used for the OPLS analysis.

In the input models, disgust sensitivity was represented by vector Y. Steroid levels and related variables, namely, maternal age, maternal pre-pregnancy BMI (calculated based on a self-report of the participant’s pre-pregnancy weight and height), pregnancy length in the first and third trimester (in days), maternal weight gain (in kg), parity (0: primipara; 1: multipara), fetus sex (0: female; 1: male), maternal diabetes and hypertension (0: no; 1: yes), and maternal pre-pregnancy smoking (1: no; 2: only occasionally; 3: yes), constituted matrix **X**. Separate OPLS models were developed for the associations measured in the first trimester, in the third trimester, and also for the changes between the first and third trimester (association between Δ scores of disgust sensitivity and Δ(steroids) calculated as the levels of steroids/disgust scores in the third trimester minus the levels of steroids/disgust scores in the first trimester). In all cases, separate OPLS models were also developed for the Pathogen disgust measured via the TDDS, the overall DS-R score, and the three individual DS-R subscales. By using this approach, one predictive component was extracted for each model. Non-homogeneities were eliminated after checking the data homogeneity in predictors using Hotelling’s statistic.

## 5. Conclusions

While in the first trimester, disgust increases with rising maternal estrogen levels and C19 Δ^5^ steroids action, which are associated with catabolism of C19 estrogen precursors, in the third trimester, more positive correlations dominate between disgust levels and steroids, reflecting both maternal immune activity and potential symptoms accompanying anxiety or other psychiatric disorders. Specifically in the first trimester, disgust sensitivity was positively associated with the presence of some active sex steroids, such as testosterone and some estrogens, and the presence of some steroids indicating progesterone and sex steroid production. This has further implications for associations between disgust sensitivity and immune system activity as these steroids are known to have immunomodulatory effects. These results are in line with the compensatory prophylaxis hypothesis, which states that disgust sensitivity is elevated when immunosuppression is higher. This adaptive mechanism is especially important during pregnancy as it is necessary to protect the mother and developing fetus from pathogens by enhancing aversion to potentially harmful substances and environments. In that regard, the results in the third trimester showed a positive association between disgust sensitivity and cortisol and progesterone metabolites, with cortisol and progesterone being known to be immunosuppressive, as well as metabolites of C19 Δ^5^ androstanes and estrogen, which are known for modulating the immune response and for the suppression of autoimmunity.

Moreover, results from the third trimester—specifically, positive associations between disgust sensitivity and cortisol and DHEA and negative associations between contamination-related disgust sensitivity and testosterone—suggest a maladaptive occurrence of elevated disgust in this period of pregnancy, possibly related to various mental disorders or their symptoms. Regarding changes in steroid levels and disgust sensitivity during pregnancy, our results showed that lower steroidogenic activity in the first trimester is associated with an increase in disgust sensitivity between the first and third trimesters. This increase is due to the heightened production of certain steroids such as androstenediol, androsterone, androstenediol sulfate, 5α-dihydrotestosterone, DHEA sulfate, and estrogen sulfates.

Understanding the neural, cognitive, and behavioral intricacies of the disgust system, together with their associations with physiological mechanisms, not only broadens our comprehension of human adaptive mechanisms but also has implications for a variety of fields, ranging from psychology and neuroscience to public health. Especially crucial is the understanding of these relationships during pregnancy, which represents a sensitive period for the mother and can significantly influence the future development of her child. Knowing how these relationships work can lead to better screening for issues related to immunity or mental health during pregnancy.

## Figures and Tables

**Figure 1 ijms-25-06857-f001:**
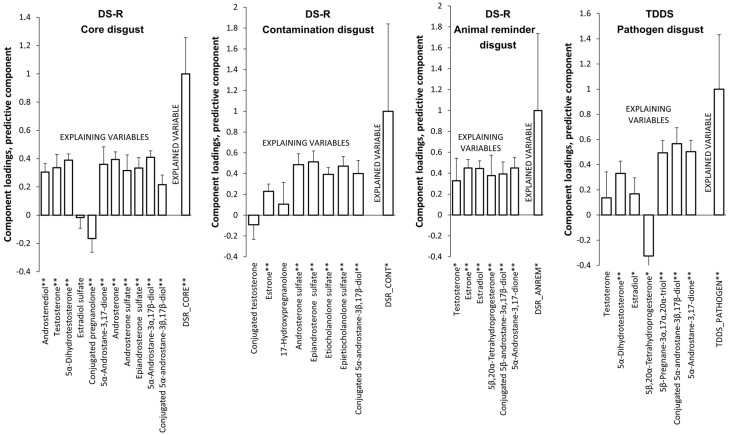
Associations between disgust sensitivity and predictors evaluated via an OPLS model in the first trimester of pregnancy. * *p* < 0.05, ** *p* < 0.01.

**Figure 2 ijms-25-06857-f002:**
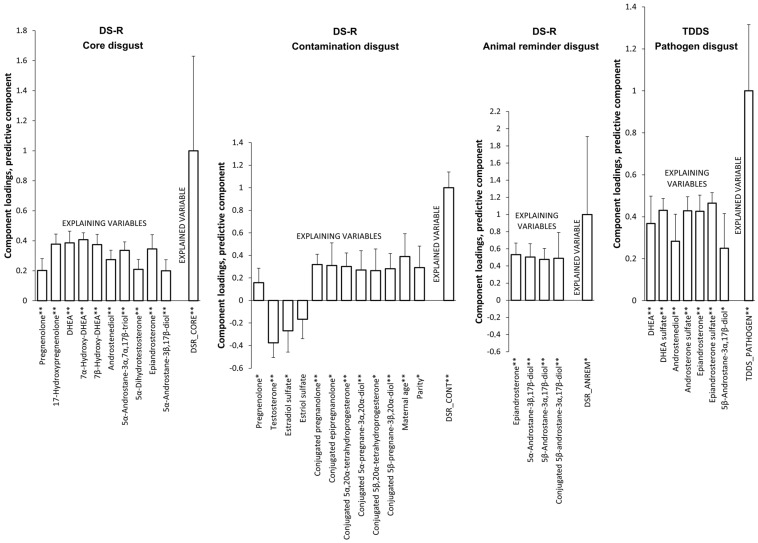
Associations between disgust sensitivity and predictors evaluated via an OPLS model in the third trimester of pregnancy. * *p* < 0.05, ** *p* < 0.01.

**Table 1 ijms-25-06857-t001:** The serum steroid levels of pregnant women in the first and third trimesters.

Steroids (nM)	Trimester 1 Median (Quartiles)	Trimester 3 Median (Quartiles)	Trimester 3 − Trimester 1 Median (Quartiles)	*p*-Value
**C21 Δ^5^ Steroids**				
Pregnenolone	4.28 (2.86, 6.22)	4.84 (3.46, 7.32)	0.563 (−0.812, 2.99)	<0.001
Pregnenolone sulfate	180 (124, 259)	212 (161, 295)	32.6 (−14.1, 84.4)	<0.001
20α-Dihydropregnenolone	4.78 (3.55, 6.09)	3.29 (2.66, 4.32)	−1.23 (−2.44, −0.327)	<0.001
20α-Dihydropregnenolone sulfate	954 (639, 1390)	849 (620, 1100)	−125 (−378, 92.5)	<0.001
17-Hydroxypregnenolone sulfate	4.6 (3.07, 7.9)	2.69 (2.01, 3.98)	−2.04 (−4.86, −0.636)	<0.001
17-Hydroxypregnenolone	7.29 (4.63, 12)	7.39 (5.05, 11.5)	0.144 (−3.54, 3.08)	0.704
16α-Hydroxypregnenolone	0.462 (0.298, 0.68)	0.806 (0.589, 1.13)	0.344 (0.0694, 0.593)	<0.001
**C19 Δ^5^ Steroids**				
Dehydroepiandrosterone (DHEA)	9.25 (6.01, 12.9)	4.63 (3.29, 6.87)	−4.12 (−6.63, −1.66)	<0.001
DHEA sulfate	2900 (1800, 4360)	1150 (666, 1810)	−1670 (−2560, −910)	<0.001
7α-Hydroxy-DHEA	0.909 (0.539, 1.43)	0.344 (0.228, 0.526)	−0.533 (−0.973, −0.233)	<0.001
7-oxo-DHEA	0.503 (0.306, 0.842)	0.519 (0.41, 0.71)	0.0141 (−0.307, 0.235)	0.647
7β-Hydroxy-DHEA	0.598 (0.342, 0.885)	0.285 (0.175, 0.382)	−0.334 (−0.565, −0.124)	<0.001
Androstenediol	2.72 (1.85, 4.1)	1.13 (0.741, 2.15)	−1.24 (−2.16, −0.51)	<0.001
Androstenediol sulfate	322 (194, 512)	174 (112, 261)	−130 (−307, −25.3)	<0.001
5-Androstene-3β,7α,17β-triol	0.459 (0.281, 0.729)	0.123 (0.084, 0.233)	−0.305 (−0.571, −0.165)	<0.001
5-Androstene-3β,7β,17β-triol	0.509 (0.319, 0.873)	0.231 (0.144, 0.319)	−0.312 (−0.572, −0.134)	<0.001
5-Androstene-3β,16α,17β-triol	0.206 (0.131, 0.319)	0.496 (0.337, 0.788)	0.266 (0.107, 0.536)	<0.001
5-Androstene-3β,16α,17β-triol sulfate	32.6 (18.9, 61.6)	78.2 (46.9, 139)	34 (10.5, 78.2)	<0.001
**C21 Δ^4^ Steroids**				
Progesterone	104 (76.5, 135)	382 (281, 489)	277 (183, 390)	<0.001
20α-Dihydroprogesterone	22.5 (17.2, 29.2)	49.2 (36.3, 67.7)	26.3 (15.6, 43)	<0.001
17-Hydroxyprogesterone	11 (8.47, 14.1)	17.4 (13.8, 21.3)	5.7 (1.69, 10.9)	<0.001
16α-Hydroxyprogesterone	2.14 (1.57, 2.79)	10.2 (7.62, 14.3)	8.07 (5.31, 12.2)	<0.001
17α,20α-Dihydroxy-4-pregnene-3-one	3.08 (2.18, 4.12)	4.44 (3.34, 5.85)	1.25 (0.332, 2.54)	<0.001
**C19 Δ^4^ Steroids**				
Androstenedione	7.22 (5.44, 9.79)	6.62 (4.8, 9.26)	−0.966 (−2.54, 0.667)	<0.001
Testosterone	2.9 (2.1, 3.9)	2.54 (1.69, 4.17)	−0.245 (−0.805, 0.784)	0.308
Conjugated testosterone	1.08 (0.407, 2.39)	4.2 (2.12, 7.29)	2.1 (0.335, 5.67)	<0.001
5α-Dihydrotestosterone	0.903 (0.596, 1.45)	0.443 (0.27, 0.682)	−0.446 (−0.761, −0.167)	<0.001
Conjugated 5α-dihydrotestosterone	1.42 (0.854, 2.09)	1 (0.677, 1.58)	−0.288 (−1.09, 0.364)	<0.001
**Estrogens**				
Estrone	3.57 (1.99, 6.85)	17 (10.4, 31.4)	13.4 (5.26, 29.9)	<0.001
Estrone sulfate	5.67 (3, 11.7)	27.3 (16.8, 47.6)	21.7 (9.41, 38.2)	<0.001
Estradiol	3.31 (1.55, 5.77)	31 (18.4, 48.5)	27 (16.2, 44.4)	<0.001
Estradiol sulfate	3.98 (2.18, 6.83)	19.8 (10.3, 35.2)	15.8 (4.91, 29.6)	<0.001
Estriol	1.89 (0.726, 4.44)	74.8 (40.8, 119)	74.3 (40.9, 116)	<0.001
**C21 5α/β-reduced Steroids**				
5α-Dihydroprogesterone	10.4 (7.09, 13.7)	65.1 (42.8, 92.5)	54.3 (30.7, 80.6)	<0.001
Allopregnanolone	6.71 (5.05, 9.09)	31.1 (21.1, 42.7)	23.8 (14.8, 34)	<0.001
Allopregnanolone sulfate	128 (82.7, 227)	1220 (739, 1870)	1080 (534, 1690)	<0.001
Isopregnanolone	2.05 (1.38, 2.86)	7.96 (4.44, 11.9)	5.28 (2.73, 9.51)	<0.001
Isopregnanolone sulfate	92.3 (59, 143)	605 (344, 958)	477 (263, 837)	<0.001
5β-Dihydroprogesterone	0.123 (0.043, 0.235)	2.02 (1.1, 3.26)	1.89 (0.978, 3.01)	<0.001
Pregnanolone	1.91 (1.06, 3.06)	22.8 (14.5, 29.9)	20 (13.5, 27.1)	<0.001
Conjugated pregnanolone	76 (54, 131)	696 (480, 950)	549 (361, 806)	<0.001
Epipregnanolone	0.267 (0.137, 0.538)	1.36 (0.823, 1.97)	0.935 (0.596, 1.53)	<0.001
Conjugated epipregnanolone	23.7 (14.7, 37.3)	154 (95.4, 255)	117 (67.8, 207)	<0.001
17-Hydroxyallopregnanolone	0.11 (0.04, 0.197)	0.279 (0.124, 0.511)	0.149 (0.019, 0.314)	<0.001
17-Hydroxyallopregnanolone sulfate	6.6 (3.82, 11)	11.1 (6.83, 19.5)	3.48 (0.727, 9.93)	<0.001
17-Hydroxypregnanolone	0.195 (0.103, 0.36)	0.882 (0.595, 1.19)	0.623 (0.388, 0.899)	<0.001
Conjugated 17α-hydroxypregnanolone	24.1 (16.8, 39.6)	46.4 (31, 70.6)	19.4 (8.95, 35.1)	<0.001
5α,20α-Tetrahydroprogesterone	6.04 (4.42, 8.2)	21.7 (15.1, 31.1)	15.3 (9.37, 22.2)	<0.001
Conjugated 5α,20α-tetrahydroprogesterone	5.35 (3.3, 8.59)	15.5 (10.7, 25.2)	10.1 (4.74, 18.5)	<0.001
5α-Pregnane-3α,20α-diol	7.23 (4.83, 9.68)	23.1 (16, 32.4)	14.2 (8.53, 23.9)	<0.001
Conjugated 5α-pregnane-3α,20α-diol	1650 (1100, 2580)	6550 (4220, 9420)	4440 (2660, 7260)	<0.001
5α-Pregnane-3β,20α-diol	2.24 (1.57, 3.27)	6.59 (4.35, 10.5)	3.65 (2.33, 7.26)	<0.001
Conjugated 5α-pregnane-3β,20α-diol	3720 (2150, 6210)	10,900 (7600, 17,000)	6550 (3540, 11,500)	<0.001
5β,20α-Tetrahydroprogesterone	0.096 (0.062, 0.171)	1.96 (1.4, 2.66)	1.8 (1.28, 2.47)	<0.001
Conjugated 5β,20α-tetrahydroprogesterone	2.65 (1.68, 4.06)	7.48 (5.02, 9.64)	4.22 (2.05, 6.78)	<0.001
5β-Pregnane-3α,20α-diol	0.829 (0.584, 1.21)	6.61 (5.06, 8.53)	5.53 (4.19, 7.16)	<0.001
Conjugated 5β-pregnane-3α,20α-diol	297 (229, 427)	1070 (794, 1460)	750 (498, 998)	<0.001
5β-Pregnane-3β,20α-diol	0.18 (0.113, 0.313)	0.593 (0.396, 0.927)	0.393 (0.209, 0.665)	<0.001
Conjugated 5β-pregnane-3β,20α-diol	189 (127, 286)	644 (461, 920)	430 (253, 624)	<0.001
5α-Pregnane-3α,17α,20α-triol	0.276 (0.146, 0.474)	0.179 (0.104, 0.321)	−0.082 (−0.199, 0.006)	<0.001
Conjugated 5α-pregnane-3α,17α,20α-triol	28.5 (10.5, 58)	35.5 (11.8, 87.5)	3.2 (−5.57, 23.6)	0.001
5β-Pregnane-3α,17α,20α-triol	2.8 (1.88, 3.94)	4.79 (3.16, 6.17)	1.52 (0.35, 2.82)	<0.001
Conjugated 5β-pregnane-3α,17α,20α-triol	116 (82.2, 216)	162 (117, 269)	34.8 (−21.4, 102)	<0.001
5α-Androstane-3,17-dione	0.521 (0.34, 0.744)	0.425 (0.297, 0.625)	−0.0559 (−0.197, 0.0546)	<0.001
**C19 5α/β-reduced Steroids**				
Androsterone	0.897 (0.705, 1.21)	0.568 (0.427, 0.815)	−0.336 (−0.549, −0.12)	<0.001
Androsterone sulfate	1290 (718, 2100)	594 (333, 986)	−540 (−1250, −258)	<0.001
Epiandrosterone	0.333 (0.205, 0.497)	0.158 (0.102, 0.251)	−0.16 (−0.288, −0.062)	<0.001
Epiandrosterone sulfate	348 (214, 496)	129 (76.8, 187)	−199 (−312, −116)	<0.001
Etiocholanolone	0.265 (0.164, 0.421)	0.233 (0.148, 0.396)	−0.025 (−0.103, 0.062)	0.103
Etiocholanolone sulfate	53.5 (34.2, 84.6)	36.6 (20, 56.2)	−15.6 (−36.2, −4.48)	<0.001
Epietiocholanolone sulfate	17 (10.4, 35.5)	9.45 (6.06, 17.8)	−7.24 (−18.5, −2.09)	<0.001
5α-Androstane-3α,17β-diol	0.174 (0.118, 0.245)	0.079 (0.061, 0.11)	−0.09 (−0.145, −0.05)	<0.001
Conjugated 5α-androstane-3α,17β-diol	19.7 (14.2, 28.7)	13.2 (8.46, 18)	−7.12 (−14.7, −1.3)	<0.001
5α-Androstane-3β,17β-diol	0.064 (0.024, 0.141)	0.032 (0.01, 0.075	−0.022 (−0.081, −0.002)	<0.001
Conjugated 5α-androstane-3β,17β-diol	26.6 (16.1, 56.5)	12.5 (7.12, 23.6)	−14.8 (−35.1, −4.69)	<0.001
5β-Androstane-3α,17β-diol	0.011 (0.006, 0.018)	0.007 (0.003, 0.013)	−0.004 (−0.009, 0.002)	<0.001
Conjugated 5β-androstane-3α,17β-diol	3.98 (2.5, 6.59)	2.85 (1.78, 4.9)	−0.916 (−2.09, 0.272)	<0.001
**Corticoids and 11β-hydroxy-androstanes**				
Cortisol	389 (308, 473)	632 (537, 824)	231 (148, 386)	<0.001
Cortisone	106 (77.1, 148)	170 (128, 252)	52.8 (22.8, 111)	<0.001
Corticosterone	12.5 (7.38, 18.6)	19.5 (14.6, 27.2)	6.96 (−0.918, 14.7)	<0.001
11-Deoxycortisol	0.74 (0.168, 1.87)	3.49 (1.06, 7.02)	2.57 (0.423, 5.22)	<0.001
21-Deoxycortisol	0.081 (0.029, 0.243)	0.117 (0.0656, 0.259)	0.029 (−0.025, 0.098)	<0.001
3α,5α-Tetrahydrocorticosterone	0.039 (0.0173, 0.087)	0.028 (0.010, 0.056)	−0.009 (−0.043, 0.003)	<0.001
3α,5β-Tetrahydrocorticosterone	0.124 (0.0409, 0.328)	0.06 (0.022, 0.127)	−0.052 (−0.228, 0.0112)	<0.001
11β-Hydroxyandrostenedione	48.3 (29.9, 74.9)	60 (39.5, 102)	8.58 (−1.77, 30)	<0.001
11β-Hydroxyandrosterone	1.2 (0.702, 2.14)	0.316 (0.188, 0.528)	−0.897 (−1.73, −0.432)	<0.001
11β-Hydroxyandrosterone sulfate	12.2 (8.5, 18.3)	7.45 (4.59, 11.6)	−4.56 (−9.27, −0.811)	<0.001
11β-Hydroxyepiandrosterone	0.048 (0.023, 0.101)	0.012 (0.005, 0.025)	−0.031 (−0.074, −0.009)	<0.001
11β-Hydroxyepiandrosterone sulfate	0.783 (0.413, 1.25)	1.81 (0.938, 3.18)	0.974 (0.203, 1.94)	<0.001
11β-Hydroxyetiocholanolone	0.927 (0.576, 1.39)	0.422 (0.241, 0.673)	−0.44 (−0.799, −0.213)	<0.001
11β-Hydroxyetiocholanolone sulfate	3.01 (1.82, 4.85)	1.77 (1.05, 2.93)	−1.06 (−2.17, −0.174)	<0.001

Note: *p*-values show significant differences between trimesters.

**Table 2 ijms-25-06857-t002:** Descriptive statistics for the DS-R (Core, Animal reminder, Contamination disgust subscale scores and the overall DS-R score) and the Pathogen domain of the TDDS in the first (T1) and third (T3) trimester.

Questionnaire	*n*	Mean	Median (Quartiles)	SD	Min.	Max.	Cronbach’s Alpha
Overall DS-R T1	169	56.2	55 (45, 66)	14.3	21	93	0.843
Overall DS-R T3	176	55.1	56 (44.8, 66)	15.3	15	91	0.874
Core T1	169	29.4	30 (25, 35)	6.96	11	48	0.676
Core T3	176	28.4	28 (23, 34)	7.56	8	47	0.763
Contamination T1	169	8.53	8 (6, 11)	3.62	2	19	0.562
Contamination T3	176	8.53	9 (6,11)	3.70	0	19	0.622
Animal reminder T1	169	18.3	18 (14, 23)	6.37	3	32	0.760
Animal reminder T3	176	18.2	19 (13, 23)	6.66	0	32	0.792
Pathogen TDDS T1	174	23.3	24 (17, 28)	7.35	4	40	0.734
Pathogen TDDS T3	176	23.4	23 (18, 29)	7.70	7	41	0.798

Note: The table shows the total number of participants who filled out each questionnaire at each time point.

**Table 3 ijms-25-06857-t003:** Associations between disgust sensitivity measured via the DS-R and predictors evaluated via an OPLS model and multiple regression in the first trimester of pregnancy.

	OPLS (Predictive Component)							Multiple Regression		
Variable	Variable Importance	*t*-Statistics		Component Loading	*t*-Statistics	R *^a^*		Regression Coefficient	*t*-Statistics	
**DS-R, Overall score**										
Androstenediol	0.988	2.91	*	0.289	8.96	0.559	**	0.045	3.32	**
Estrone	0.939	3.24	**	−0.062	−2.43	−0.114	*	−0.042	−2.15	*
17-Hydroxypregnanolone	1.245	2.88	*	0.225	3.18	0.434	**	0.056	2.72	*
5β-Pregnane-3α,17α,20α-triol	0.789	2.04	*	0.228	4.58	0.441	**	0.036	2.60	*
5α-Androstane-3,17-dione	1.025	4.26	**	0.361	11.28	0.697	**	0.046	3.35	**
Androsterone	1.096	3.30	**	0.388	9.55	0.750	**	0.049	3.13	**
Androsterone sulfate	0.981	3.68	**	0.310	11.08	0.595	**	0.044	2.21	*
Epiandrosterone	1.010	2.81	*	0.389	7.71	0.752	**	0.046	1.95	*
Epiandrosterone sulfate	1.092	5.81	**	0.332	7.09	0.640	**	0.049	3.21	**
5α-Androstane-3α,17β-diol	0.822	2.76	*	0.354	8.29	0.686	**	0.037	4.73	**
5α-Androstane-3β,17β-diol	1.007	2.34	*	0.308	10.12	0.595	**	0.045	2.08	*
5β-Androstane-3α,17β-diol	1.082	2.35	*	0.194	4.04	0.369	**	0.049	2.09	*
3α,5β-Tetrahydrocorticosterone	0.825	2.57	*	0.165	3.05	0.320	**	0.037	1.66	
DS-R, Overall score, trimester 1				1.000	2.91	0.313	*			
**Explained variability**	9.8% (6.8% after cross−validation)									
**DS-R, Core disgust**										
Androstenediol	0.998	2.34	*	0.305	9.63	0.610	**	0.051	2.14	*
Testosterone	1.199	4.01	**	0.336	6.80	0.672	**	0.061	3.96	**
5α-Dihydrotestosterone	0.974	3.16	**	0.389	16.63	0.780	**	0.050	2.78	*
Estradiol sulfate	0.709	2.36	*	−0.017	−0.41	−0.028		−0.036	−2.56	*
Conjugated pregnanolone	0.598	1.94	*	−0.166	−3.24	−0.327	**	−0.031	−2.06	*
5α-Androstane-3,17-dione	1.110	3.52	**	0.359	5.49	0.719	**	0.057	3.46	**
Androsterone	0.923	8.32	**	0.394	13.88	0.788	**	0.047	7.34	**
Androsterone sulfate	0.988	3.28	**	0.315	5.40	0.632	**	0.051	3.18	**
Epiandrosterone sulfate	0.944	3.92	**	0.334	8.53	0.670	**	0.048	3.79	**
5α-Androstane-3α,17β-diol	1.139	4.40	**	0.409	16.91	0.820	**	0.058	4.31	**
Conjugated 5α-androstane-3β,17β-diol	1.230	6.41	**	0.216	6.00	0.423	**	0.063	6.28	**
DS-R, Core disgust, trimester 1				1.000	7.40	0.338	**			
**Explained variability**	11.4% (9.7% after cross−validation)									
**DS-R, Contamination disgust**										
Conjugated testosterone	0.820	2.78	*	−0.091	−1.22	−0.145		−0.051	−2.06	*
Estrone	0.810	6.88	**	0.230	6.33	0.373	**	0.051	3.54	**
17-Hydroxypregnanolone	0.856	2.34	*	0.106	0.96	0.170		0.053	3.10	**
Androsterone sulfate	1.214	2.76	*	0.487	8.88	0.789	**	0.076	2.31	*
Epiandrosterone sulfate	1.221	3.39	**	0.513	9.21	0.832	**	0.076	2.55	*
Etiocholanolone sulfate	0.788	2.31	*	0.393	11.47	0.640	**	0.049	1.99	*
Epietiocholanolone sulfate	1.202	2.91	*	0.472	9.85	0.751	**	0.075	2.60	*
Conjugated 5α-androstane-3β,17β-diol	0.953	2.40	*	0.401	6.13	0.654	**	0.060	1.75	
DS-R, Contamination disgust, trimester 1				1.000	2.25	0.286	*			
**Explained variability**	8.2% (5.2% after cross−validation)									
**DS-R, Animal reminder disgust**										
estosterone	0.752	2.24	*	0.328	2.90	0.505	*	0.061	1.89	
Estrone	1.072	4.66	**	0.450	10.64	0.723	**	0.087	5.98	**
Estradiol	1.243	6.54	**	0.444	11.46	0.676	**	0.100	4.63	**
5β,20α-Tetrahydroprogesterone	0.833	2.02	*	0.376	3.65	0.576	**	0.067	2.14	*
Conjugated 5β-androstane-3α,17β-diol	0.940	3.08	**	0.393	6.55	0.605	**	0.076	2.89	*
5α-Androstane-3,17-dione	1.078	3.89	**	0.449	8.42	0.690	**	0.087	3.03	**
DS-R, Animal reminder disgust, trimester 1				1.000	2.57	0.303	*			
**Explained variability**	9.2% (7.4% after cross-validation)									

^a^ R. Component loadings expressed as correlation coefficients with predictive component, * *p* < 0.05, ** *p* < 0.01.

**Table 4 ijms-25-06857-t004:** Associations between Pathogen disgust measured via the TDDS and predictors evaluated via an OPLS model and multiple regression in the first trimester of pregnancy.

	OPLS (Predictive Component)							Multiple Regression		
Variable	Variable Importance	*t*-Statistics		Component Loading	*t*-Statistics	R *^a^*		Regression Coefficient	*t*-Statistics	
Testosterone	0.521	1.94	*	0.137	1.27	0.195		0.056	1.71	
5α-Dihydrotestosterone	0.896	3.30	**	0.331	6.40	0.468	**	0.097	2.45	*
Estradiol	0.766	2.72	*	0.168	2.53	0.231	*	0.083	2.66	*
5β,20α-Tetrahydroprogesterone	1.127	3.50	**	−0.326	−2.76	−0.462	*	−0.122	−3.41	**
5β-Pregnane-3α,17α,20α-triol	0.792	2.11	*	0.494	9.74	0.698	**	0.086	1.98	*
Conjugated 5α-androstane-3β,17β-diol	1.565	5.93	**	0.567	8.51	0.800	**	0.169	6.74	**
5α-Androstane-3,17-dione	0.996	6.58	**	0.504	10.66	0.712	**	0.108	4.69	**
TDDS, Pathogen disgust, trimester 1				1.000	4.38	0.407	**			
**Explained variability**	16.6% (13.5% after cross-validation)									

^a^ R. Component loadings expressed as correlation coefficients with predictive component, * *p* < 0.05, ** *p* < 0.01.

**Table 5 ijms-25-06857-t005:** Associations between disgust sensitivity measured via the DS-R and predictors evaluated via an OPLS model and multiple regression in the third trimester of pregnancy.

	OPLS (Predictive Component)							Multiple Regression		
Variable	Variable Importance	*t*-Statistics		Component Loading	*t*-Statistics	R *^a^*		Regression Coefficient	*t*-Statistics	
**DS-R, Overall score**										
7α-Hydroxy-DHEA	0.757	1.96	*	0.390	8.96	0.626	**	0.091	2.18	*
Androstenediol	0.856	3.01	**	0.356	6.13	0.572	**	0.152	2.69	*
5β-Androstane-3α,17β-diol	1.213	8.34	**	0.401	12.35	0.641	**	0.164	2.37	*
Cortisol	0.961	4.38	**	0.364	7.18	0.584	**	0.137	3.89	**
DS-R, Overall score, trimester 3				1.000	4.30	0.415	**			
**Explained variability**	11.7% (8.8% after cross-validation)									
**DS-R, Core disgust**										
Pregnenolone	0.785	2.65	*	0.202	4.88	0.419	**	0.041	1.85	
17-Hydroxypregnenolone	1.361	4.82	**	0.377	10.59	0.782	**	0.071	5.51	**
DHEA	0.924	3.14	**	0.386	9.49	0.801	**	0.048	2.66	*
7α-Hydroxy-DHEA	1.153	5.17	**	0.407	16.53	0.844	**	0.060	3.29	**
7β-Hydroxy-DHEA	1.030	4.15	**	0.374	10.36	0.775	**	0.054	2.56	*
Androstenediol	1.125	5.00	**	0.275	8.36	0.569	**	0.059	3.68	**
5-Androstene-3 β,7α,17β-triol	0.984	2.89	*	0.336	11.14	0.696	**	0.051	1.97	*
5α-Dihydrotestosterone	0.566	2.21	*	0.210	6.00	0.434	**	0.029	2.20	*
Epiandrosterone	1.053	3.40	**	0.345	6.85	0.716	**	0.055	6.77	**
5α-Androstane-3β,17β-diol	0.792	2.45	*	0.200	5.18	0.415	**	0.041	1.76	
DS-R, Core disgust, trimester 3				1.000	3.01	0.342	**			
**Explained variability**	11.7% (8.8% after cross-validation)									
**DS-R, Contamination disgust**										
Pregnenolone	0.690	3.19	**	0.158	2.33	0.222	*	0.105	2.24	*
Testosterone	1.048	4.14	**	−0.376	−5.44	−0.546	**	−0.151	−4.06	**
Estradiol sulfate	0.950	2.59	*	−0.269	−2.70	−0.393	*	−0.158	−2.06	*
Estriol sulfate	0.636	2.53	*	−0.167	−1.85	−0.255		−0.134	−2.69	*
Conjugated pregnanolone	1.032	5.42	**	0.318	6.55	0.445	**	0.035	0.90	
Conjugated epipregnanolone	1.026	2.79	*	0.310	2.92	0.431	*	0.052	1.08	
Conjugated 5α,20α-tetrahydroprogesterone	1.140	5.96	**	0.301	4.73	0.410	**	0.095	3.07	**
Conjugated 5α-pregnane-3α,20α-diol	1.131	3.01	**	0.270	3.00	0.368	**	0.103	1.32	
Conjugated 5β,20α-tetrahydroprogesterone	0.978	2.60	*	0.265	2.62	0.362	*	0.080	0.95	
Conjugated 5β-pregnane-3β,20α-diol	0.877	4.29	**	0.282	3.94	0.386	**	0.025	0.73	
Maternal age	1.357	3.74	**	0.390	3.65	0.569	**	0.237	3.60	**
Multipara (1: yes/0: no)	0.925	3.14	**	0.292	2.89	0.428	*	0.185	5.37	**
DS-R, Contamination disgust, trimester 3				1.000	13.56	0.570	**			
**Explained variability**	32.5% (23.1% after cross-validation)									
**DS-R, Animal reminder disgust**										
Epiandrosterone	1.080	4.83	**	0.532	7.50	0.676	**	0.135	4.59	**
5α-Androstane-3β,17β-diol	0.894	2.93	*	0.505	6.19	0.641	**	0.112	2.34	*
5β-Androstane-3α,17β-diol	0.951	4.60	**	0.477	7.10	0.605	**	0.119	3.54	**
Conjugated 5β-androstane-3α,17β-diol	1.063	2.98	*	0.489	3.08	0.613	**	0.133	2.50	*
DS-R, Animal reminder disgust, trimester 3				1.000	2.09	0.317	*			
**Explained variability**	10% (7.8% after cross-validation)									

^a^ R. Component loadings expressed as correlation coefficients with predictive component, * *p* < 0.05, ** *p* < 0.01.

**Table 6 ijms-25-06857-t006:** Associations between Pathogen disgust measured via the TDDS and predictors evaluated via an OPLS model and multiple regression in the third trimester of pregnancy.

	OPLS (Predictive Component)							Multiple Regression		
Variable	Variable Importance	*t*-Statistics		Component Loading	*t*-Statistics	R *^a^*		Regression Coefficient	*t*-Statistics	
DHEA	0.737	3.04	**	0.368	5.37	0.667	**	0.056	2.72	*
DHEA sulfate	0.907	3.42	**	0.431	14.40	0.783	**	0.068	3.21	**
Androstenediol	0.533	1.93	*	0.283	4.15	0.513	**	0.040	2.46	*
Androsterone sulfate	1.116	6.79	**	0.428	12.16	0.778	**	0.084	5.63	**
Epiandrosterone	1.383	5.35	**	0.426	10.60	0.773	**	0.104	5.53	**
Epiandrosterone sulfate	1.129	5.90	**	0.465	17.83	0.845	**	0.085	3.84	**
5β-Androstane-3α,17β-diol	0.957	2.90	*	0.250	2.87	0.453	*	0.072	2.17	*
TDDS, Pathogen disgust, trimester 3				1.000	6.00	0.361	**			
**Explained variability**	13% (11.1% after cross-validation)									

^a^ R. Component loadings expressed as correlation coefficients with predictive component, * *p* < 0.05, ** *p* < 0.01.

**Table 7 ijms-25-06857-t007:** Associations between Δdisgust (trimester 3−trimester 1) measured via the DS-R and predictors in the first trimester and Δpredictors evaluated via an OPLS model and multiple regression in pregnancy.

	OPLS (Predictive Component)							Multiple Regression		
Variable	Variable Importance	*t*-Statistics		Component Loading	*t*-Statistics	R *^a^*		Regression Coefficient	*t*-Statistics	
**ΔDS-R, Overall score**										
Pregnenolone sulfate	0.771	2.97	*	−0.180	−8.45	−0.527	**	−0.021	−3.14	**
17-Hydroxypregnenolone sulfate	0.617	2.77	*	−0.162	−5.63	−0.470	**	−0.017	−2.69	*
16α-Hydroxypregnenolone	0.764	3.25	**	−0.194	−6.32	−0.569	**	−0.021	−3.56	**
DHEA sulfate	1.080	2.44	*	−0.207	−5.54	−0.605	**	−0.030	−2.85	*
7α-Hydroxy-DHEA	0.793	2.73	*	−0.237	−5.40	−0.693	**	−0.022	−3.93	**
7α-oxo-DHEA	0.642	2.71	*	−0.152	−10.42	−0.444	**	−0.018	−2.29	*
Androstenediol	0.641	3.62	**	−0.185	−6.52	−0.541	**	−0.018	−3.16	**
Androstenediol sulfate	0.852	4.59	**	−0.122	−3.62	−0.355	**	−0.023	−4.11	**
20α-Dihydroprogesterone	0.664	1.98	*	−0.092	−2.22	−0.268	*	−0.018	−1.53	
17-Hydroxyprogesterone	0.955	4.46	**	−0.146	−4.67	−0.428	**	−0.026	−3.39	**
16α-Hydroxyprogesterone	1.179	6.49	**	−0.144	−4.17	−0.420	**	−0.032	−3.91	**
17,20α-Dihydroxy-4-pregnen-3-one	0.919	3.25	**	−0.179	−6.10	−0.527	**	−0.025	−2.61	*
Androstenedione	1.266	6.07	**	−0.227	−9.73	−0.663	**	−0.035	−3.75	**
Testosterone	1.416	9.51	**	−0.232	−8.69	−0.680	**	−0.039	−4.74	**
5α-Dihydrotestosterone	1.164	5.27	**	−0.244	−17.51	−0.712	**	−0.032	−4.52	**
Estrone	0.718	3.21	**	−0.094	−4.67	−0.265	**	−0.020	−4.78	**
Estradiol	0.779	2.92	*	−0.105	−4.24	−0.304	**	−0.021	−2.62	*
17α-Hydroxyallopregnanolone	1.242	3.34	**	−0.167	−4.75	−0.490	**	−0.034	−2.33	*
17α-Hydroxypregnanolone	1.305	3.71	**	−0.125	−2.41	−0.366	*	−0.036	−2.35	*
5β-Pregnane-3α,17,20α-triol	0.970	3.31	**	−0.140	−4.81	−0.412	**	−0.027	−2.44	*
5α-Androstane-3,17-dione	1.437	4.69	**	−0.233	−6.53	−0.682	**	−0.039	−3.48	**
Androsterone	1.552	12.86	**	−0.275	−15.44	−0.804	**	−0.042	−4.75	**
Androsterone sulfate	0.799	2.57	*	−0.134	−4.14	−0.391	**	−0.022	−2.89	*
Epiandrosterone	1.295	6.09	**	−0.260	−9.38	−0.761	**	−0.035	−4.87	**
Epiandrosterone sulfate	0.808	1.93	*	−0.151	−5.84	−0.441	**	−0.022	−2.19	*
5α-Androstane-3α,17β-diol	1.619	12.04	**	−0.260	−14.96	−0.762	**	−0.044	−6.40	**
5α-Androstane-3β,17β-diol	1.184	5.87	**	−0.192	−10.04	−0.561	**	−0.032	−7.80	**
Maternal age	0.692	2.67	*	0.125	3.89	0.364	**	0.019	2.08	*
Male sex of the fetus	0.790	2.16	*	−0.035	−1.26	−0.103		−0.022	−2.45	*
DS-R Overall score	0.851	3.18	**	−0.066	−2.62	−0.194	*	−0.023	−2.64	*
Pregnancy length	0.564	2.04	*	0.051	2.65	0.148	*	0.015	2.27	*
ΔDHEA sulfate	1.128	2.11	*	0.194	5.72	0.566	**	0.031	2.27	*
ΔAndrostenediol	0.857	2.49	*	0.156	4.55	0.458	**	0.023	2.07	*
Δ5α-Dihydrotestosterone	0.980	4.69	**	0.149	5.41	0.437	**	0.027	5.27	**
ΔAndrosterone	0.754	3.00	*	0.142	8.92	0.418	**	0.021	2.54	*
Δ5α-Androstane-3β,17β-diol	0.848	2.06	*	0.104	5.15	0.303	**	0.023	2.39	*
ΔAndrostenediol sulfate	0.608	2.34	*	0.010	0.31	0.029		0.017	2.31	*
ΔDS-R, Overall score				1.000	11.93	0.486	**			
**Explained variability**	23.7% (19.3% after cross-validation)									
**ΔDS-R, Core disgust**										
17-Hydroxyprogesterone	0.914	2.49	*	−0.257	−2.99	−0.519	*	−0.044	−1.82	
16α-Hydroxyprogesterone	0.771	3.95	**	−0.203	−2.47	−0.410	*	−0.037	−2.45	*
Androstenedione	0.929	2.58	*	−0.361	−7.89	−0.729	**	−0.045	−2.12	*
Testosterone	0.886	11.65	**	−0.349	−14.24	−0.702	**	−0.043	−4.23	**
17-Hydroxypregnanolone	1.238	2.91	*	−0.220	−2.24	−0.446	*	−0.060	−2.07	*
5α-Androstane-3,17-dione	0.793	4.33	**	−0.359	−16.89	−0.724	**	−0.038	−2.82	*
Androsterone	0.862	4.60	**	−0.373	−7.41	−0.752	**	−0.042	−3.36	**
Androsterone sulfate	1.379	5.09	**	−0.288	−3.92	−0.573	**	−0.067	−7.75	**
Epiandrosterone sulfate	1.194	4.75	**	−0.295	−3.48	−0.587	**	−0.058	−10.59	**
Epietiocholanolone sulfate	0.769	2.34	*	−0.227	−2.75	−0.450	*	−0.037	−2.89	*
5α-Androstane-3α,17β-diol	1.068	2.74	*	−0.342	−6.91	−0.689	**	−0.052	−2.78	*
5α-Androstane-3β,17β-diol	0.839	4.42	**	−0.273	−4.22	−0.550	**	−0.041	−6.07	**
DS-R, Overall score	1.121	3.30	**	−0.154	−2.42	−0.312	*	−0.054	−3.09	**
ΔDS-R, Core disgust				1.000	2.80	0.352	*			
**Explained variability**	12.4% (9.9% after cross-validation)									
**ΔDS-R, Contamination disgust**										
Pregnenolone sulfate	0.695	2.53	*	−0.190	−10.55	−0.427	**	−0.031	−2.05	*
20α-Dihydropregnenolone	0.868	4.04	**	−0.145	−5.26	−0.325	**	−0.038	−4.67	**
DHEA sulfate	0.730	2.68	*	−0.205	−5.05	−0.460	**	−0.032	−3.03	**
Androstenediol	0.872	4.48	**	−0.207	−8.62	−0.468	**	−0.039	−5.64	**
Androstenediol sulfate	0.912	3.64	**	−0.141	−3.66	−0.316	**	−0.040	−2.66	*
Androstenedione	1.171	2.79	*	−0.280	−7.71	−0.633	**	−0.052	−3.31	**
Testosterone	1.041	2.11	*	−0.278	−4.95	−0.628	**	−0.046	−2.26	*
Estrone	0.935	2.73	*	−0.186	−5.90	−0.399	**	−0.041	−2.63	*
Allopregnanolone	0.942	4.42	**	−0.172	−4.80	−0.389	**	−0.042	−4.26	**
Allopregnanolone sulfate	0.709	2.18	*	−0.073	−1.89	−0.164		−0.031	−1.85	
Isopregnanolone sulfate	0.785	1.90	*	−0.104	−2.76	−0.233	*	−0.035	−1.59	
17α-Hydroxypregnanolone	1.275	2.97	*	−0.199	−5.27	−0.448	**	−0.056	−3.48	**
5α,20α-Tetrahydroprogesterone	0.914	4.62	**	−0.164	−3.81	−0.372	**	−0.040	−3.60	**
5α-Pregnane-3α,20α-diol	0.632	4.27	**	−0.150	−4.82	−0.338	**	−0.028	−5.52	**
5β-Pregnane-3α,20α-diol	0.733	2.60	*	−0.150	−4.56	−0.340	**	−0.032	−2.89	*
5β-Pregnane-3β,20α-diol	0.900	4.13	**	−0.130	−3.59	−0.279	**	−0.040	−4.20	**
5β-Pregnane-3α,17α,20α-triol	0.967	2.73	*	−0.203	−5.01	−0.457	**	−0.043	−2.90	*
5α-Androstane-3,17-dione	1.274	5.03	**	−0.318	−13.20	−0.720	**	−0.056	−3.40	**
Androsterone	1.222	5.01	**	−0.319	−9.19	−0.720	**	−0.054	−5.36	**
Androsterone sulfate	1.384	6.74	**	−0.246	−6.37	−0.553	**	−0.061	−5.98	**
Epiandrosterone	0.761	3.80	**	−0.269	−17.69	−0.609	**	−0.034	−2.80	*
Epiandrosterone sulfate	1.181	4.83	**	−0.246	−8.24	−0.551	**	−0.052	−3.88	**
Etiocholanolone	0.818	2.29	*	−0.131	−2.69	−0.296	*	−0.036	−2.56	*
Conjugated 5α-androstane-3α,17β-diol	0.870	3.59	**	−0.151	−3.41	−0.336	**	−0.039	−3.15	**
Conjugated 5α-androstane-3β,17β-diol	0.874	2.10	*	−0.136	−3.15	−0.301	**	−0.039	−1.84	
Cortisone	1.256	3.77	**	−0.202	−4.02	−0.456	**	−0.056	−2.65	*
Maternal age	0.965	3.43	**	0.153	5.17	0.344	**	0.043	3.14	**
Pregnancy length	1.142	3.74	**	−0.077	−1.15	−0.172		−0.051	−3.04	**
ΔAndrostenediol sulfate	1.477	18.03	**	−0.133	−6.36	−0.298	**	−0.065	−7.19	**
ΔPregnancy length	0.982	3.15	**	0.081	2.37	0.182	*	0.043	2.69	*
ΔDS-R, Contamination disgust				1.000	11.03	0.545	**			
**Explained variability**	29.7% (24.9% after cross-validation)									
**Δ** **DS-R, Animal reminder disgust**										
7-oxo-DHEA	0.837	2.81	*	−0.165	−4.43	−0.423	**	−0.029	−3.41	**
20α-Dihydroprogesterone	1.210	3.93	**	−0.238	−4.74	−0.608	**	−0.042	−4.37	**
16α-Hydroxyprogesterone	1.150	5.03	**	−0.248	−6.41	−0.634	**	−0.040	−5.40	**
17α,20α-Dihydroxy-4-pregnene-3-one	0.851	2.17	*	−0.239	−6.31	−0.612	**	−0.030	−2.95	*
Testosterone	0.990	3.34	**	−0.271	−13.71	−0.692	**	−0.035	−3.99	**
5α-Dihydrotestosterone	0.979	4.33	**	−0.258	−5.93	−0.660	**	−0.034	−3.38	**
Estradiol	0.844	2.62	*	−0.164	−3.95	−0.414	**	−0.030	−2.50	*
17α-Hydroxyallopregnanolone	0.898	4.74	**	−0.263	−11.02	−0.674	**	−0.031	−3.03	**
17α-Hydroxypregnanolone	0.981	3.14	**	−0.210	−5.96	−0.539	**	−0.034	−2.82	*
3α,5α-Tetrahydroprogsterone	1.042	2.61	*	−0.250	−8.76	−0.639	**	−0.036	−2.59	*
5α-Pregnane-3α,20α-diol	0.840	2.34	*	−0.248	−5.20	−0.635	**	−0.029	−2.52	*
3α,5β-Tetrahydroprogsterone	1.446	3.89	**	−0.204	−7.26	−0.521	**	−0.051	−2.67	*
5β-Pregnane-3α,20α-diol	1.146	3.78	**	−0.204	−12.24	−0.521	**	−0.040	−2.69	*
5β-Pregnane-3α,17α,20α-triol	0.774	4.00	**	−0.212	−8.23	−0.545	**	−0.027	−3.78	**
Androsterone	0.972	11.39	**	−0.273	−7.07	−0.699	**	−0.034	−6.20	**
Epiandrosterone	1.023	3.51	**	−0.215	−4.07	−0.550	**	−0.036	−2.94	*
5α-Androstane-3α,17β-diol	1.231	3.52	**	−0.262	−6.79	−0.670	**	−0.043	−2.91	*
5α-Androstane-3β,17β-diol	1.148	1.93	*	−0.217	−3.69	−0.554	**	−0.040	−1.79	
Parity	0.908	3.68	**	0.081	1.60	0.206		0.032	2.84	*
DS-R, Animal reminder disgust	0.944	3.23	**	−0.080	−2.67	−0.205	*	−0.033	−2.36	*
Pregnancy length	0.716	2.65	*	0.114	3.14	0.292	**	0.025	3.18	**
Δ5α-Dihydrotestosterone	0.714	3.04	**	0.204	5.83	0.523	**	0.025	2.76	*
Δ5α-Androstane-3α,17β-diol	1.013	3.00	**	0.186	9.32	0.477	**	0.035	2.32	*
ΔPregnancy length	0.993	2.67	*	0.102	2.13	0.261	*	0.035	3.09	**
ΔDS-R, Animal reminder disgust				1.000	20.08	0.438	**			
**Explained variability**	19.2% (15.3% after cross-validation)									

^a^ R. Component loadings expressed as correlation coefficients with predictive component; * *p* < 0.05, ** *p* < 0.01.

**Table 8 ijms-25-06857-t008:** Associations between ΔPathogen disgust (trimester 3−trimester 1) measured via the TDDS and predictors in the first trimester and Δpredictors evaluated via an OPLS model and multiple regression in pregnancy.

	OPLS, Predictive Component							Multiple Regression		
Variable	Variable Importance	*t*-Statistics		Component Loading	*t*-Statistics	R *^a^*		Regression Coefficient	*t*-Statistics	
Pregnenolone	0.642	2.83	*	0.183	1.98	0.259	*	0.075	1.98	*
TDDS_PATHOGEN	1.001	4.05	**	−0.333	−4.28	−0.474	**	−0.118	−4.83	**
ΔEstrone sulfate	1.140	3.91	**	0.517	10.61	0.740	**	0.134	2.92	*
ΔEstradiol sulfate	1.061	3.80	**	0.566	16.62	0.804	**	0.125	2.67	*
ΔEstriol sulfate	1.077	9.86	**	0.553	14.69	0.804	**	0.127	6.58	**
ΔTDDS, Pathogen disgust				1.000	2.72	0.375	*			
**Explained variability**	14% (11.2% after cross-validation)									

^a^ R. Component loadings expressed as correlation coefficients with predictive component; * *p* < 0.05, ** *p* < 0.01.

## Data Availability

The data presented in this study are available on request from the corresponding author. The data are not publicly available due to privacy restrictions.

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
