# Peer review of "Association between Disgust Sensitivity during Pregnancy and Endogenous Steroids: A Longitudinal Study"

_ijms, 2024, doi:10.3390/ijms25136857_

Round 1

Reviewer 1 Report

Comments and Suggestions for Authors

I am pleased to have the chance to review the article "Association between Disgust Sensitivity During Pregnancy and Endogenous Steroids: A Longitudinal Study." Despite its interesting nature, the topic's clinical or therapeutic implications are still uncertain. Furthermore, some crucial issues require attention.

Major Remarks:

- The plagiarism check shows a 23% similarity with other papers, which is deemed inappropriate. Kindly paraphrase the research using your own language, refraining from directly quoting other sources or reusing your own work.

- The abstract and the entire paper are disorganised. The introduction lacks the study's objective. Revise it in accordance with STROBE principles and append the checklist as a supplement.

- The discussion area lacks proper organisation and is disorderly. The topic lacks suitable clinical consequences at the end. Adhering to the STROBE principles for prospective studies could enhance the manuscript's quality.

Comments on the Quality of English Language

The manuscript could benefit from revision by a native English speaker to improve clarity and readability.

Reviewer 2 Report

Comments and Suggestions for Authors

 The aim of the presented study was to assess the relationship between serum steroid concentrations and disgust. The tested steroids included the groups C21 Δ5 Steroids, C21 Δ4 Steroids, C19 Δ4 Steroids, C21 5α/β-reduced Steroids, Corticoids and 11β-hydroxy-an-drostanes, which constituted a total of approximately 90 steroids. Disgust was assessed on the basis of test questionnaires on a sample of 179 in the first and third trimester pregnant women.

After careful reading of the manuscript, I find that the summary, introduction and other chapters exhaustively and adequately reflect the issues raised. The conclusions presented by the authors are consistent with the evidence and address the main research problem. In addition, I believe that the description of the research methodology and the selection of bibliographic items are up-to-date and correct (out of 74 bibliographic items, 15 papers have been published since 2019).

I recommend the publication in its current form.

Comments on the Quality of English Language

-

Author Response

Thank you for your kind feedback and for recognizing the strengths of our study.

In the revised version of the manuscript, we incorporated comments from other reviewers. We hope that these changes have significantly improved our manuscript.

Reviewer 3 Report

Comments and Suggestions for Authors

The authors explored relationships between serum steroid content in trimester 1 and trimester 3 of pregnancy and disgust. The manuscript is well-written, and no plagiarism is detected. However, lines from 433 to 464 could be carefully rewritten as the identity with literature available data is high in these two paragraphs. 

The authors found significant changes in levels of most detected steroids levels compared to trimester 1 and trimester 3 of pregnancy. The statistically significant relationships were found for levels of steroids in trimester 1 and trimester 3 as well as changes of steroids levels and disgust scores. The obtained results spread current knowledge about the impact of levels of steroids on disgust sensitivity.

The introduction section could be shortened. The results of the mentioned studies could not be described in so much detail.  

The results are presented in Tables, but it could be clearer if some results were presented in Figures. As 91 molecules were detected in serum (56 unconjugated steroids and 35 polar conjugates of steroids) and their correlations with disgust scores obtained using questionnaires were analyzed it was not easy clearly to present the obtained results, but several Figures could be prepared especially as the figure does not need to be black and white. 

The conclusion section should be carefully rewritten focusing on obtained results. 

Several minor suggestions

Line 33-36 These two sentences should be carefully rewritten, their meanings may not be clear to readers at present form

Line 64-68 Please carefully rewrite the sentence 

Line 84 Please consider replacing term one with the author or it was found

The Title of Table 1 could be modified as there are no characteristics of participants presented in it, but steroid concentrations

Kind regards
